# Prediction modelling studies for medical usage rates in mass gatherings: A systematic review

Hans Van Remoortel[1]*, Hans Scheers[1], Emmy De Buck[1,2,3], Winne Haenen[4], Philippe Vandekerckhove[2,5]

1 Centre for Evidence-Based Practice, Belgian Red Cross, Mechelen, Belgium, 2 Department of Public Health and Primary Care, Faculty of Medicine, KU Leuven, Leuven, Belgium, 3 Cochrane First Aid, Mechelen, Belgium, 4 Federal Public Service Health, Food Chain Safety and Environment, Brussels, Belgium, 5 Belgian Red Cross, Mechelen, Belgium

* hans.vanremoortel@cebap.org

**Data Availability Statement:** All relevant data are within the manuscript and its Supporting Information files.

## Abstract

### Background

Mass gathering manifestations attended by large crowds are an increasingly common feature of society. In parallel, an increased number of studies have been conducted that developed and/or validated a model to predict medical usage rates at these manifestations.

### Aims

To conduct a systematic review to screen, analyse and critically appraise those studies that developed or validated a multivariable statistical model to predict medical usage rates at mass gatherings. To identify those biomedical, psychosocial and environmental predictors that are associated with increased medical usage rates and to summarise the predictive performance of the models.

### Method

We searched for relevant prediction modelling studies in six databases. The predictors from multivariable regression models were listed for each medical usage rate outcome (i.e. patient presentation rate (PPR), transfer to hospital rate (TTHR) and the incidence of new injuries). The GRADE methodology (Grades of Recommendation, Assessment, Development and Evaluation) was used to assess the certainty of evidence.

### Results

We identified 7,036 references and finally included 16 prediction models which were developed (n = 13) or validated (n = 3) in the USA (n = 8), Australia (n = 4), Japan (n = 1), Singapore (n = 1), South Africa (n = 1) and The Netherlands (n = 1), with a combined audience of >48 million people in >1700 mass gatherings. Variables to predict medical usage rates were biomedical (i.e. age, gender, level of competition, training characteristics and type of injury) and environmental predictors (i.e. crowd size, accommodation, weather, free water

**Funding:** This work was made possible through funding from the Foundation for Scientific Research of the Belgian Red Cross. One of the activities of the Belgian Red Cross is providing first aid training to laypeople.

**Competing interests:** The authors have declared that no competing interests exist.

availability, time of the manifestation and type of the manifestation) (low-certainty evidence). Evidence from 3 studies indicated that using Arbon's or Zeitz' model in other contexts significantly over- or underestimated medical usage rates (from 22% overestimation to 81% underestimation).

## Conclusions

This systematic review identified multivariable models with biomedical and environmental predictors for medical usage rates at mass gatherings. Since the overall certainty of the evidence is low and the predictive performance is generally poor, proper development and validation of a context-specific model is recommended.

## Introduction

A mass gathering has been defined by the World Health Organization (WHO) as an occasion, either organized or spontaneous, where the number of people attending is sufficient to strain the planning and response resources of the community, city, or nation hosting the manifestation [1].

Since mass gatherings attended by large crowds have become a more frequent feature of society, mass gathering medicine was highlighted as a new discipline at the World Health Assembly of Ministers of Health in Geneva in May 2014 [2]. As a consequence, the amount of international initiatives and meetings on mass gathering medicine has increased over the past decade as has the number of experts and the amount of publications on pre-event planning and surveillance for mass gathering. Mass gatherings are associated with increased health risks and hazards such as the transmission of communicable diseases, exacerbation of non-communicable diseases and comorbidities (e.g. diabetes, hypertension, COPD, cardiovascular events) and an impact on mental or physical health and psychosocial disorders [3]. Furthermore, the mental health consequences of traumatic incidents at mass gatherings can be prolonged with stress to people, families, and communities resulting in short-term fear of death as well as general distress, anxiety, excessive alcohol consumption, and other psychiatric disorders. If mass gatherings are improperly managed, this can lead to human, material, economic or environmental losses and impacts [4]. Therefore, the development of (cost-)effective methods for the planning and handling of the health risks associated with mass gatherings will strengthen global health security, prevent excessive emergency health problems and associated economic loss, and mitigate potential societal disruption in host and home communities [5].

To have a better understanding of the health effects of mass gatherings, a conceptual model for mass gathering health care was published in 2004 by Paul Arbon [6]. This model divided the key characteristics of mass gathering manifestations into three interrelated domains that may have an impact on the Patient Presentation Rate (PPR), the Transport To Hospital Rate (TTHR) and the level and extent of healthcare services: 1) the biomedical domain (i.e. biomedical influences such as demographic characteristics of the audience), 2) the psychosocial domain (i.e. psychological and social influences within mass gatherings including individual and crowd behaviour) and 3) the environmental domain (i.e. environmental features of a mass gathering including terrain and climatological conditions). Although most scientific papers on mass gathering are descriptive, i.e. without proper statistical analysis to predict medical usage rates, recently more prediction modelling studies have been developed and/or validated to have a better understanding of the patient care required at such manifestations. In order to

formulate evidence-based, robust and effective interventions in the planning and management of mass gatherings, the scientific underpinning of Arbon's conceptual model by a systematic screening, analysis and critical appraisal of prediction modelling studies for medical usage rates at mass gatherings was needed.

This systematic review aimed to identify multivariable prediction models for medical usage rates at mass gatherings, to summarize evidence for individual biomedical, psychosocial and environmental predictors at mass gatherings, and to summarise the predictive performance of these models.

## Material and methods

### Protocol and registration

We carried out a systematic literature review according to a predefined protocol, which was not registered beforehand [7]. We planned and reported the systematic review in accordance with the Preferred Reporting Items for Systematic Reviews and Meta-Analyses (PRISMA checklist, S1 File) [8].

### Eligibility criteria

Studies were eligible for inclusion if they answered the following PICO (Population, Intervention, Comparison, Outcome) question: "Which predictive models (I) are available for emergency services planning (O) during mass gathering manifestations (P)?" Full texts of potentially relevant articles were reviewed according to the following inclusion and exclusion criteria:

- Population: studies performed on all types of mass gatherings were included, such as sport (spectator) manifestations, (indoor/outdoor) music concerts and/or festivals. A mass gathering has been defined by the World Health Organization (WHO) as an occasion, either organized or spontaneous, where the number of people attending is sufficient to strain the planning and response resources of the community, city, or nation hosting the manifestation [1].

- Intervention/Predictors: we included studies that described a multivariable statistical model and extracted data of the predictors. Multivariable models represent a more realistic picture, rather than looking at a single variable (univariate associations) and they provide a powerful test of significance compared to univariate techniques. We included studies that had the intention to evaluate more than one predictor variable in a multivariable model, regardless of how many predictor variables remained in the final model. Evacuation models, opinion-based or theory-based models, and statistical models based on univariate (correlation) analysis were excluded.

- Outcome: we included medical usage rates such as Patient Presentation Rate (PPR), the Transport To Hospital Rate (TTHR) or the incidence of new injuries.

- Study design: prediction model development studies without external validation, prediction model development studies with external application in few independent mass gatherings or validation on an extensive list of independent mass gatherings (i.e. big data analysis), external model validation studies or studies that applied observations from few mass gatherings to another prediction model, were included according to the Checklist for critical Appraisal and data extraction for systematic Reviews of prediction Modelling Studies (CHARMS) [9].

- Language: no language restrictions were applied.

## Data sources and searches

Eligible studies were identified by searching the following databases: MEDLINE (via the PubMed interface), Embase (via Embase.com), the Cochrane Library, CINAHL, Web of Science and Scopus from the time of inception of the database until 14 May 2019. We developed search strategies for each database using index terms and free text terms (S2 File). Search yields were exported to a citation program (EndNote X7.5) and duplicates were discarded.

## Study selection

Two reviewers (HVR and HS) independently screened the titles and abstracts of all references yielded by the search. Subsequently, the full text of each article that potentially met the eligibility criteria was obtained, and after a full-text assessment, studies that did not meet the selection criteria were excluded. Any discrepancies between reviewers were resolved by consensus or by consulting a third reviewer (EDB). For each included study, the reference lists and first 20 related citations in PubMed were screened for additional relevant records.

## Data extraction

Data concerning study design (type of prediction modelling study), study aims and hypothesis, population characteristics (participation eligibility and recruitment method; participation description; details of mass gathering manifestations; study dates), candidate predictors (dichotomous/categorical/continuous variables), outcome measures (medical usage rates), effect sizes, statistical model, and study quality were extracted independently by the two reviewers.

## Risk of bias assessment

The PROBAST (Prediction model Risk Of Bias ASsessment Tool) checklist items were used to assess the risk of bias and concerns for applicability for each study [10]. These items include 20 signalling questions across 4 domains: participants, predictors, outcome and analysis. Signalling questions were answered as 'yes', 'probably yes', 'no', 'probably no' or 'no information' and risk of bias was assessed for each domain. A domain where all signalling questions were answered as (probably) yes was judged as 'low risk of bias'. An answer of (probably) no on 1 or more questions indicated the potential for bias, whereas no information indicated insufficient information. The risk of bias assessment was performed by the two reviewers independently.

## Data synthesis

**Individual predictors for medical usage rates.** The predictors (both the statistically significant ($p<0.05$) and statistically non-significant ones) from multivariable statistical models were pooled into different categories for each type of mass gathering manifestation (music concert, spectator sport manifestation, sport manifestation, mixed manifestation (sport, music, public exhibition)) corresponding to the three main domains for mass gathering health according to Arbon's conceptual model: biomedical domain, psychosocial domain, environmental domain [6]. The direction of the association between the candidate predictors and the outcome variables was expressed as positive (e.g. night manifestations are associated with higher patient presentation rates compared to day manifestations) or negative (e.g. free water availability is associated with lower patient presentation rates).

**Predictive performance of the models.** Predictive accuracy measures of the models, such as the $R^2$ or the mean/median error, were extracted and summarized. The $R^2$ is the square of the correlation and measures the proportion of variation in the dependent variable (i.e.

medical usage rates) that can be attributed to the independent variable (i.e. predictor variables). The $R^2$ indicates how well the regression model fits the observed data, ranging from 0% (no fit) to 100% (perfect fit). The predictive performance is considered as very weak ($R^2$ of 0–4%), weak ($R^2$ of 4 to 16%), moderate ($R^2$ of 16 to 36%), strong ($R^2$ of 36 to 64%) or very strong ($R^2$ of 64% to 100%) [11].

Information on type of mass gathering, outcomes measured and the model that was validated with the data collected (i.e. the reference model), was summarized. Results were reported as a % underestimation or % overestimation (compared to the reference model).

## Grading of the evidence

The GRADE approach (Grading of Recommendations, Assessment, Development and Evaluation) was used to assess the certainty of the evidence (also known as quality of evidence or confidence in effect estimates) [12]. Since no meta-analyses were possible, we used the GRADE guidelines for rating the certainty in evidence in the absence of a single estimate of effect [13]. The certainty of the evidence was graded as 'high' (further research is very unlikely to change our confidence in the effect estimate), 'moderate' (further research is likely to have an important impact on our confidence in the effect estimate), 'low' (further research is very likely to have an important impact on our confidence in the effect estimate and is likely to change the estimate) or 'very low' (any estimate of effect is very uncertain). The initial certainty level of the included prediction modelling studies was set at 'high' because the association between the predictors and outcomes was considered irrespective of any causal connection. Eight criteria were considered to further downgrade or upgrade the certainty of the evidence: five criteria who might potentially downgrade the overall certainty of the evidence (i.e. methodological limitations of the study, indirectness, imprecision, inconsistency and likelihood of publication bias) and three criteria who might potentially upgrade the overall certainty of the evidence (i.e. large effect, dose-response relation in the effect, and opposing plausible residual bias or confounding). Methodological limitations of the studies were assessed by considering the overall risk of bias judgement across studies based on the risk of bias assessment of the 4 PROBAST domains (i.e. participants, predictors, outcome and analysis). Indirectness was assessed by making a global judgement on how dissimilar the research evidence is to the PICO question at hand (in terms of population, interventions and outcomes across studies). The PROBAST tool was used to identify concerns regarding the applicability of each included study (i.e. when the populations, predictors or outcomes of the study differ from those specified in the review question) and an overall judgement across studies was made. Imprecision was assessed by considering the optimal information size (or the total number of events for binary outcomes and the number of participants in continuous outcomes) across all studies. A threshold of 400 or less is concerning for imprecision [14]. Results may also be imprecise when the 95% confidence intervals of all studies or of the largest studies include no effect and clinically meaningful benefits or harms. A global judgement on inconsistency was done by evaluating the consistency of the direction and primarily the difference in the magnitude of association between the predictor variables and the outcomes across studies (since statistical measures of heterogeneity were not available). Widely differing estimates of the effects indicated inconsistency. Publication bias was suspected when the body of evidence consisted of only small positive studies or when studies were reported in trial registries but not published.

A large magnitude of effect (i.e. large association between the predictor variable and outcome) was considered in case the relative risk or odds ratio is 2–5 or 0.5–0.2 with no plausible confounders in the majority of studies. Since this review was not focused on drugs or pharmaceutical agents, assessing a dose-response gradient was not applicable here. Finally, we only

included studies that described a multivariable statistical model. Therefore, making a judgement whether all plausible confounders and biases from the prediction modelling studies unaccounted for in the adjusted/multivariate analyses (i.e. all residual confounders) and may lead to an underestimated association is not applicable here.

The two reviewers independently rated the certainty of the evidence for each outcome. Any discrepancies between reviewers were resolved by consensus or by consulting a third reviewer (EDB).

## Results

### Study selection

The systematic literature search resulted in a total of 7,036 citations (after removing duplicates), which were screened by two reviewers independently. Fig 1 represents the study selection flowchart. We included 16 studies that developed (n = 13) or externally applied (n = 3) a multivariable statistical model to predict medical usage rates in mass gathering manifestations. No studies were identified that externally validated prediction models against a big data set of mass gatherings.

### Study characteristics

A total amount of >1,700 mass gathering manifestations (median[range]: 2.5[1–405] mass gatherings per study) attended by >48 million people were included to develop and/or validate these models. A mix of different types of mass gathering manifestations were included such as sports (spectator) manifestations (e.g. soccer games, auto races, (half-)marathon, n = 12 (75%)), music concerts (indoor/outdoor, n = 8 (50%)), fete/carnivals (n = 4, 25%), public exhibitions and ceremonial manifestations (n = 3, 19%). The majority of the studies (n = 12, 75%) were conducted in the USA (n = 8) and Australia (n = 4). The other studies were performed in Japan (n = 1), Singapore (n = 1), South Africa (n = 1) and The Netherlands (n = 1). Data were collected in 2 studies between 1980–1995, in 7 studies between 1995–2005, and in 7 studies between 2005–2015. Patient influx at first aid posts, expressed as total number or rate (per 1,000 or 10,000 attendees), was the outcome of interest in most of the studies (n = 14). Other outcomes included in the prediction model were the number of transfers to hospital (per 1,000 or 10,000 attendees) (n = 7) or the incidence of new (non-)medical injuries/complications (n = 3). All studies (except one) investigated whether at least one of the following environmental candidate predictors were associated with medical usage (rates): 1) weather conditions (n = 12: average/maximal daily temperature; humidity; heat index; dew point; % sunshine; wind speed; precipitation; barometric pressure), 2) crowd size (n = 12), 3) type of the manifestation (n = 12), 4) time of the manifestation (n = 7: night vs day; duration; year of the manifestation; season; day of the week), 5) venue accommodation (n = 7: mobile vs seated; indoor vs outdoor; bounded vs unbounded; focussed vs extended; maximum venue capacity; access to venue), 6) presence of alcohol (n = 4) or 7) free water availability (n = 1). Five studies included biomedical candidate predictors into their (univariate) model: 1) demographics (n = 5: age; gender; BMI), 2) level of competition (n = 3: running experience; running pace category; competitive vs non-competitive), 3) training characteristics (amount of training; type of training) and 4) injury status (n = 1: injuries incurred in the 12 months prior to the manifestation). None of the studies included psychosocial candidate predictors (e.g. crowd behaviour, reason for attendance, length of stay) in the model. Four studies used general linear regression analysis to develop a multivariable prediction model. Other types of generalized linear regression analysis included Poisson regression analysis (n = 4), logistic regression analysis (n = 3), and negative binomial regression analysis (n = 2). One study applied non-linear regression analysis

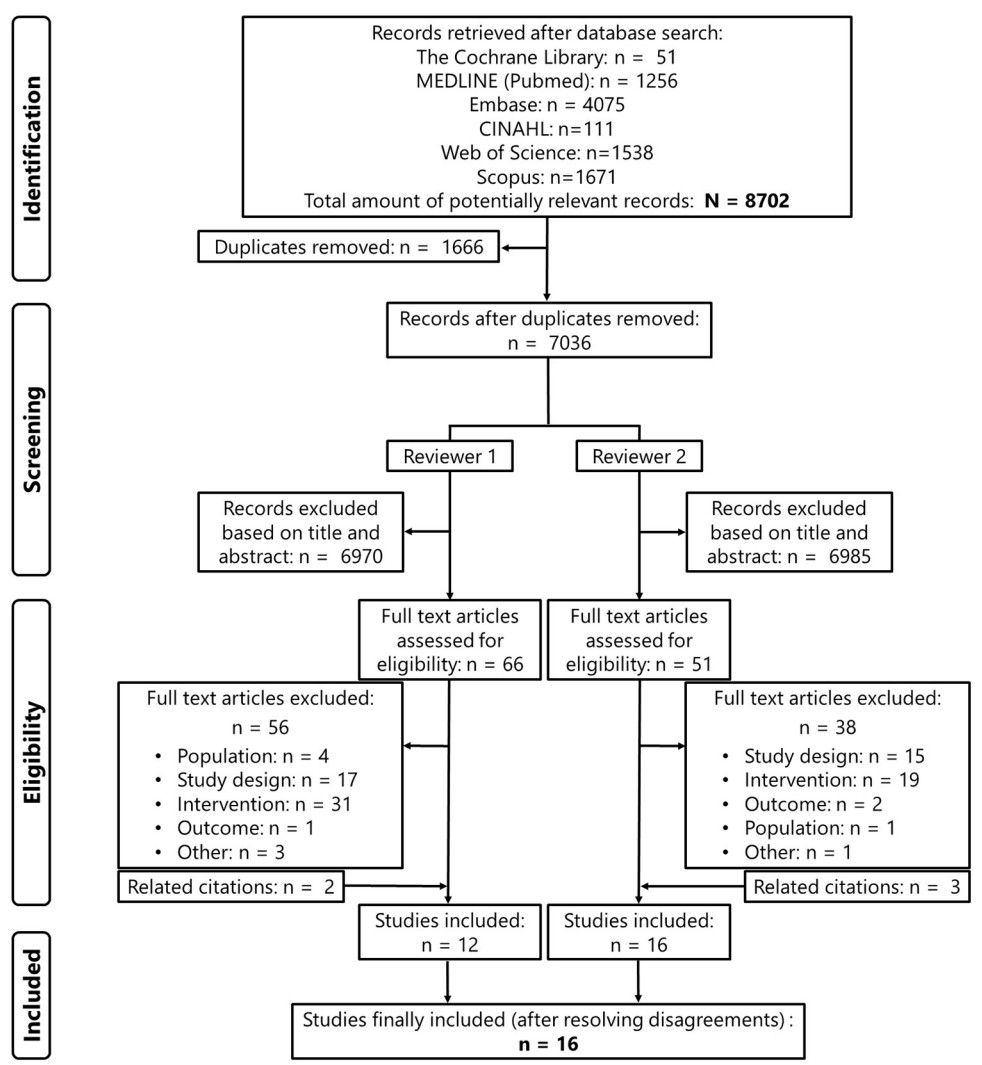

**Fig 1. Study identification and selection process of the systematic review.**

(Classification And Regression Trees (CART)). Details on the characteristics of the included studies can be found in Table 1.

## Risk of bias assessment

Individual judgements about each PROBAST risk of bias item (i.e. 20 signalling questions according to 4 domains) can be found in S1 and S2 Figs. PROBAST domains that were most prone to bias were the methods of analysis used (high risk of bias in 13 studies (81%)) and the participant recruitment (high risk of bias in 10 studies (62%)).

## Factors that predict patient presentation (rate)

Ten multivariable regression models to predict patient presentation (rate) were developed. The following predictor variables were included in these models: weather conditions (in 8 models), crowd size (in 4 models), type of the manifestation (in 8 models), venue accommodation (in 4 models), time of the manifestation (in 3 models), free water availability (in 1 model),

**Table 1. Characteristics of included studies.**

| Author, year, Country | Population—mass gathering manifestations | Outcome | Candidate predictors | Type of regression analysis |
|---|---|---|---|---|
| **Prediction model development studies without external validation** | | | | |
| Arbon, 2001, Australia [15] | Number of manifestations: 201 (mix of manifestations: bounded vs unbounded; focused vs extended) | PPR (per 1,000 attendees) | Environmental (n = 7): crowd size; venue accommodation (mobile vs seated, bounded vs unbounded, indoor vs outdoor); weather conditions (humidity; temperature); time of the manifestation (night vs day); type of the manifestation | Linear |
| | Total sample size: 12,046,436 spectators—11,956 patients | TTHR (per 1,000 attendees) | | |
| Grange, 1999, USA [16] | Number of manifestations: 405 concerts (Outdoor concerts: Blockbuster Pavilion (1993–1995) in Devore Hollywood Bowl (1991–1995) in Hollywood, Los Angeles Coliseum (1991, 1992) in Los Angeles. Indoor concerts: Los Angeles Sports Arena (1991–1994) in Los Angeles, Long Beach Arena (1991, 1993) in Long Beach) | PPR (per 10,000) | Environmental (n = 3): crowd size; weather conditions (temperature); type of the manifestation | Poisson |
| | Total sample size: 4.638.099 total attendees–1,492 total patients (mean PPR = 2.1/10,000) | | | |
| Locoh-Donou, 2016, USA [17] | Number of manifestations: 79 mass gatherings (29 athletic manifestations; 11 football games; 23 concerts; 16 public exhibitions) | PPR (per 10,000) | Environmental (n = 9): crowd size; venue accommodation (% occupied seating, bounded vs unbounded; indoor vs outdoor); weather conditions (heat index; air conditioning); free water availability; type of the manifestation; alcohol presence | Poisson |
| | Total sample size: 839,599 spectators—670 patient presentations (PPR = 7.98/10,000) | | | |
| Milsten, 2003, USA [18] | Number of manifestations: 215 mass gatherings (27 NLF football games; 168 MLB baseball games; 20 rock concerts) | PPR (per 10,000) | Environmental (n = 4): weather conditions (humidity; temperature; heat index); type of the manifestation | Poisson |
| | Total sample size: 9,633,462 spectators—5,899 patient encounters (mean PPR 6.1/10,000) | | | |
| Morimura, 2004, Japan [19] | Number of manifestations: FIFA World Cup Soccer Japan 2002 (32 soccer games) | PP (total) | Environmental (n = 6): crowd size; venue accommodation (maximum venue capacity, access to venue); weather conditions (humidity, temperature, wind velocity) | Linear |
| | | PPR (per 1,000) | | |
| | Total sample size: 1,439,052 spectators—1,661 patient presentations (mean PPR 1.21/1,000; mean TTHR 0.05/1,000) | TTH (total) | | |
| | | TTHR (per 1,000 attendees) | | |
| Schwabe, 2014, South Africa [20] | Number of manifestations: 4 editions of the Two Oceans half-marathon race in Cape Town (South Africa) (2008–2011) | New medical complications (general medical complications; postural hypotension; musculoskeletal complications; gastrointestinal complications) | Biomedical (n = 4): level of competition (running experience; running pace category); demographics (age, gender) | Poisson |
| | Total sample size: 39,511 starters (21,028 men and 18,483 women); incidence medical complications = 5.14/1,000 runners (95%CI: 4.48–5.90) | | Environmental (n = 1): time of the manifestation (year of race) | |
| Selig, 2013, USA [21] | Number of manifestations: race events during 6 weekends at Kansas Speedway | PPR (per 10,000) | Environmental (n = 6): weather conditions (humidity, temperature, barometric pressure, dew point; precipitation); type of the manifestation | Negative binomial |
| | Total sample size: not reported—1,305 patient encounters (mean PPR 13/10,000; mean TTHR 0.24/10,000) | TTHR (per 10,000) | | |
| Tan, 2014, Singapore [22] | Number of manifestations: 3 editions of the Army Half-Marathon (Singapore) | PPR (per 10,000) | Biomedical (n = 3): level of competition (competitive vs non-competitive); demographics (age, gender); | Logistic |
| | Total sample size: 99,163 participants—221 casualties (MUR from 16 to 26 (casualties/10,000)) | | Environmental (n = 1): type of the manifestation | |

(*Continued*)

**Table 1.** (Continued)

| Author, year, Country | Population—mass gathering manifestations | Outcome | Candidate predictors | Type of regression analysis |
|---|---|---|---|---|
| van Poppel, 2016, The Netherlands [23] | Number of manifestations: 1 edition of the half-marathon or marathon (Lage Landen marathon, Eindhoven 2012, The Netherlands) | New running injuries | Biomedical (n = 8): training characteristics (amount and frequency of training, type of running terrain, type of training); injury status; demographics (age, gender, BMI) | Logistic |
| | Total sample size: 614 runners (43.7±11.2y, 67.1% male, BMI 23.1±2.5) returned the baseline and follow-up questionnaire: 464 ran the half-marathon and 150 the marathon; 142 (23.1%) runners reported a total of 209 new injuries. | | | |
| Westrol, 2017, USA [24] | Number of manifestations: 403 outdoor music concerts or festivals in 11 genres: country, hard rock/heavy metal, pop, electronic dance music, hip-hop/rap, alternative rock, modern rock, classic rock, adult contemporary, classical/ symphony, variety/ other | PPR (per 1,000) | Biomedical (n = 1): type of injury (intoxication vs medical vs trauma) | Linear |
| | Total sample size: 2,399,864 attendees—4,546 patients (PPR = 18.9/10,000; 1,697 transports to hospital; TTHR = 7.1/10,000) | TTHR (per 1,000) | Environmental (n = 4): crowd size; weather conditions (heat index, precipitation); type of the manifestation | |
| Woodall, 2010, Australia [25] | 156 manifestations (sporting manifestations, fetes/carnivals, spectator sports, concerts/ raves, ceremonial manifestations), 755 patient presentations. Total number of participants is not given. | Injury status | Biomedical (n = 2): demographics (age, gender) | Logistic |
| | | | Environmental (n = 5): crowd size; time of the manifestation (season, even duration); type of the manifestation; alcohol presence | |
| **Prediction model development studies with internal validation** | | | | |
| Arbon, 2018, Australia [26] | Number of manifestations: 201 (see Arbon 2001), no information reported on the 15 included manifestations from 2015–2016. | PP (total) | Environmental (n = 10): crowd size; venue accommodation (mobile vs seated; bounded vs unbounded; indoor vs outdoor vs both; focused vs extended); | Non-linear |
| | Total sample size: see Arbon 2001, no information reported on the 15 included manifestations from 2015–2016 | TTH (total) | weather conditions (humidity; temperature); time of the manifestation (night vs day vs both); type of the manifestation; alcohol presence | |
| Bowdish, 1992, USA [27] | Number of manifestations: 7 editions of the Indianapolis 500 Mile Race events (1983–1989) | PP (total) | Environmental (n = 6): crowd size; weather conditions (humidity, temperature, dew point, % sunshine, wind speed) | Linear |
| | Total sample size: not reported—105 patients | | | |
| **Studies that applied observations from few mass gatherings to an existing prediction model** | | | | |
| Fitzgibbon, 2017, USA [28] | Number of manifestations: 3 electronic dance music festivals (Moonrise Festival, Mad Decent Block Party, SweetLife Festival) | PP (total) | Environmental (n = 8): crowd size; venue accommodation (mobile vs seated, bounded vs unbounded, indoor vs outdoor); weather conditions (humidity; temperature); time of the manifestation (night vs day); type of the manifestation | N/A |
| | Total sample size: 54,500 visitors—960 patient contacts—56 patient transports | | | |
| Nable, 2014, USA [29] | Number of manifestations: 2 editions of the Baltimore Grand Prix (2011–2012) | PP (total) | Biomedical (n = 1): demographics (age) | N/A |
| | Total sample size: 261,000 spectators—216 patient encounters | TTH (total) | Environmental (n = 9): crowd size; venue accommodation (mobile vs seated, bounded vs unbounded, indoor vs outdoor); weather conditions (humidity; temperature); time of the manifestation (night vs day); type of the manifestation; alcohol presence | |

(*Continued*)

**Table 1.** (Continued)

| Author, year, Country | Population—mass gathering manifestations | Outcome | Candidate predictors | Type of regression analysis |
|---|---|---|---|---|
| Zeitz, 2005, Australia [30] | Number of manifestations: Royal Adelaide Show | PPR (per 1,000) | Environmental (n = 9): crowd size; venue accommodation (mobile vs seated, bounded vs unbounded, indoor vs outdoor); weather conditions (humidity; temperature); time of the manifestation (night vs day, day of the week); type of the manifestation | N/A |
| | Total sample size: 622,234 visitors—1,028 casualties | TTHR (per 1,000) | | |

BMI: Body Mass Index; PP: Patient Presentation; PPR: Patient Presentation Rate; TTH: Transfers To Hospital; TTHR: Transfer To Hospital Rate; N/A: Not applicable.

demographic information (in 1 model), level of competition (in 2 models). Six studies reported the full equation of the multivariable model to predict PPR [15, 19, 22, 24, 26, 27].

Fig 2 summarizes the environmental and biomedical predictors for patient presentation rate derived from multivariable regression models.

Weather conditions were found to be a significant factor to predict patient presentation (rate): humidity [15, 26], temperature [21], heat index (i.e. a combination of air temperature and relative humidity) [18, 24] and dew point [27] were positively associated with the number or rate of patient presentation at first aid posts. In one of our included studies, temperature (i.e. <23.5°C vs ≥23.5°C and <25.5°C vs ≥25.5°C) was included in a non-linear regression tree model with a lower total number of patient presentations in case of higher temperatures [26]. One study conducted in the USA found that the presence of air conditioning in a mixture of indoor mass gatherings (sport spectator manifestations, concerts, public exhibitions) was linked to a lower patient presentation rate [17]. In five studies, the following climatological parameters were not statistical significantly associated with patient presentation (rate): humidity, %sunshine and wind speed in 1 study [27], temperature in 2 studies [16, 27] and precipitation in 3 studies [18, 21, 24].

The type of the mass gathering manifestations was a significant predictor in 7 multivariable regression models. An Australian study of 201 mixed manifestations found that non-sporting manifestations resulted in a higher PPR. Three studies focused on sports (spectator) manifestations and demonstrated that football games, but also specific outdoor music manifestations (i.e. rock concerts) resulted in higher PPR compared to baseball games. One USA study investigating 6 automobile race weekends (NASCAR, Kansas Speedway, USA) found a higher PPR during race days versus practice days. In one study that predicted PPR for a mixture of 79 mass gatherings, the type of manifestation (athletic manifestations versus football; concerts versus football; public exhibitions versus football) was not a significant predictor [17].

Three studies, conducted on data from a mixture of mass gathering manifestations, found crowd size to be positively associated with the total number of patient presentations [15, 26, 31] whereas attendance was not associated with PPR in one study [16].

Manifestations at which the audience was predominantly seated (i.e. typically large stadium concerts) demonstrated a significantly lower presentation rate compared to manifestations where spectators tended to be more mobile [15]. Outdoor manifestations have statistically significant more medical presentations compared to indoor manifestations [15, 17]. A study with a multivariable prediction model development, analysing all 32 soccer games that were played in Japan at the FIFA World Cup 2002, concluded that higher venue capacity and easier venue access were linked to a lower PPR. Conflicting evidence was found for bounded (i.e. a

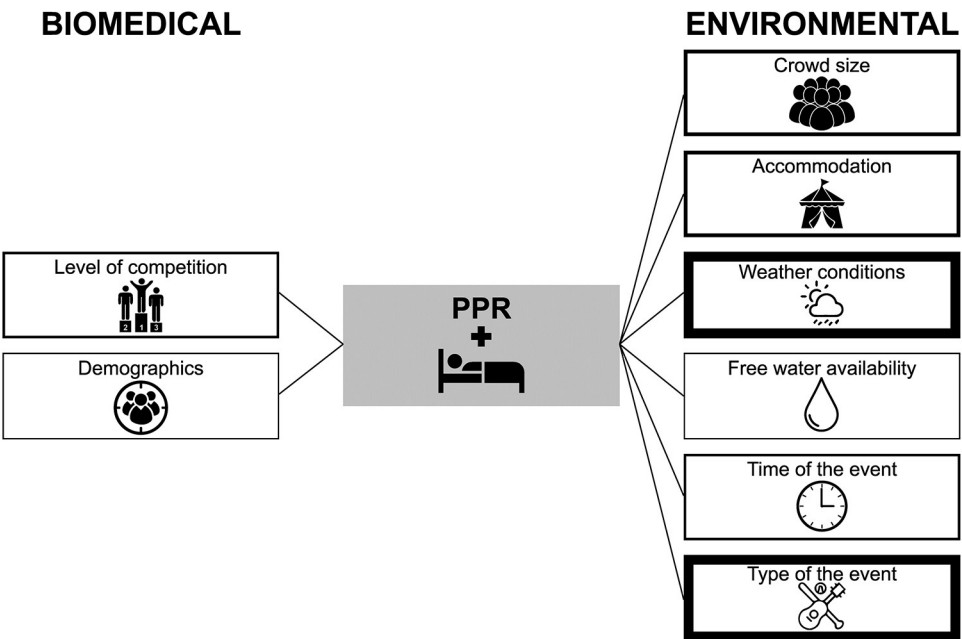

**Fig 2. Biomedical and environmental variables from multivariable regression analyses predicting the Patient Presentation Rate (PPR).** The thickness of the box represents the number of multivariable models including the following predictors: level of competition (n = 2); demographics (n = 1, age, gender); crowd size (n = 4); accommodation (n = 4, mobile vs seated, bounded vs unbounded, outdoor vs indoor, type of venue access, maximum venue capacity); weather conditions (n = 8, humidity, temperature, dew point, presence of air conditioning, % sunshine, wind speed, precipitation); free water availability (n = 1); time of the event (n = 3; day vs night, day of the week); type of the event (n = 8, music events, sport events).

manifestation contained within a boundary, often fenced) versus unbounded manifestations: one Australian model (based on data of 201 mixed manifestations) showed that bounded manifestations had a higher PPR [15] whereas one USA model (based on data of 79 mixed manifestations) showed that the unbounded manifestations resulted in a higher PPR [17]. One multivariate model to predict PPR at 405 music concerts in the USA showed that indoor versus outdoor manifestations was not a statistically significant predictor [16].

Two Australian studies found that manifestations organised during both day and night resulted in a higher PPR compared to manifestations organized during the day or night [15, 26]. One model to predict PPR at 403 music concerts found that day of the week was not a statistical significant predictor [24].

Free water availability (i.e. provided without cost to the patron) resulted in a lower PPR. In this USA model, it was shown that the absence of free water led to a two-fold increase in the PPR, even after controlling for other predictors such as weather conditions, percentage seating and alcohol availability [17].

One study found that competitiveness was positively associated with PPR during half-marathon running events, whereas the level of competition, expressed as the combination of the number of caution periods, number of lead changes, and the interval between the winner and second place, was not associated with PPR during auto race events.

Detailed information about the effect sizes of these multivariable predictors can be found in S1 Table.

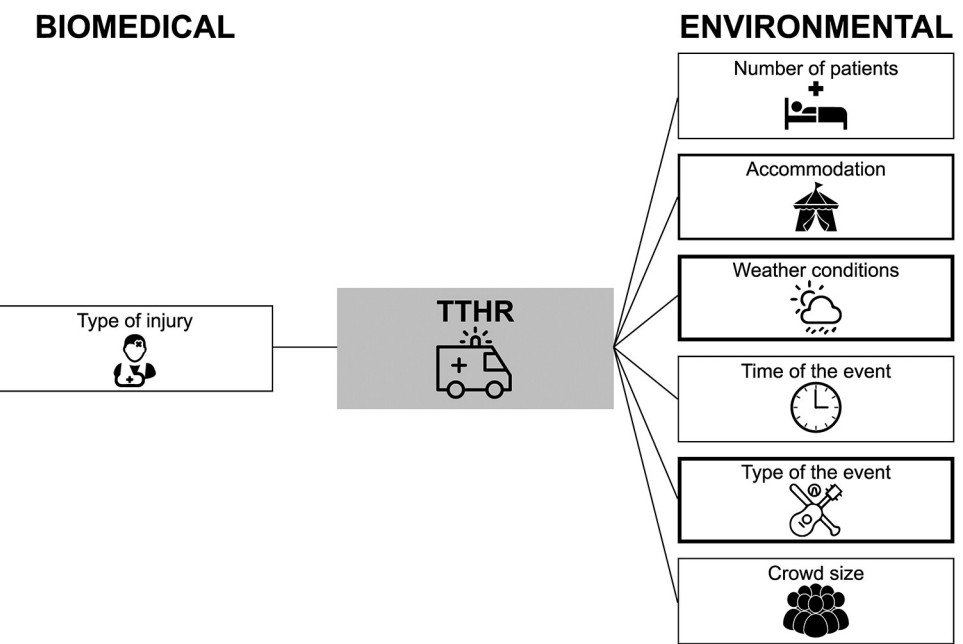

**Fig 3. Biomedical and environmental variables from multivariable regression analyses predicting The Transfer to Hospital Rate (TTHR).** The thickness of the box represents the number of multivariable models including the following predictors: type of injury (n = 1, intoxication vs medical vs trauma); number of patients (n = 1); accommodation (n = 2, mobile vs seated, bounded vs unbounded); weather conditions (n = 4, humidity, temperature, precipitation); time of the event (n = 1, day vs night); type of the event (n = 4, music genres, sport events); crowd size (n = 1).

## Factors that predict transfer to hospital (rate)

Four multivariable regression models to predict transfer to hospital (rate) were developed. The following predictor variables were included in these models: weather conditions (in 4 models), crowd size (in 1 model), venue accommodation (in 2 models), time of the manifestation (in 1 model), type of the manifestation (in 4 models), number of patient presentations (in 1 model), and type of the injury (in 1 model). Two studies reported the full equation of the multivariable model to predict TTHR [15, 26].

The biomedical and environmental multivariable factors predicting the transfer to hospital rate are depicted in Fig 3.

Humidity, temperature or the heat index ($\geq$32.2˚C) were positively associated to the TTHR [15, 24, 26]. In one multivariable regression model developed with data from auto race events, mean temperature, precipitation and type of the manifestation (practice day vs race day) were not predictive for TTHR [21]. Music genres with a significant positive association with transport rates were alternative rock and country, whereas no association was found for other music genres, music festivals versus no music festivals [24]. Manifestation type was an important predictor in the non-linear regression tree model of Arbon et al. since this predictor determined 2 decision nodes [26].

Venue accommodation was a significant predictor for transportation rates in 2 studies: more transports were predicted in case the audience was seated or bounded (compared to mobile or unbounded) [15]. A seated vs mobile audience was also included in the recent Arbon regression tree model [26]. Crowd size and number of patients evaluated were positively associated with TTHR [15, 26]. Similar to the prediction of PPR, manifestations

**BIOMEDICAL**                                                    **ENVIRONMENTAL**

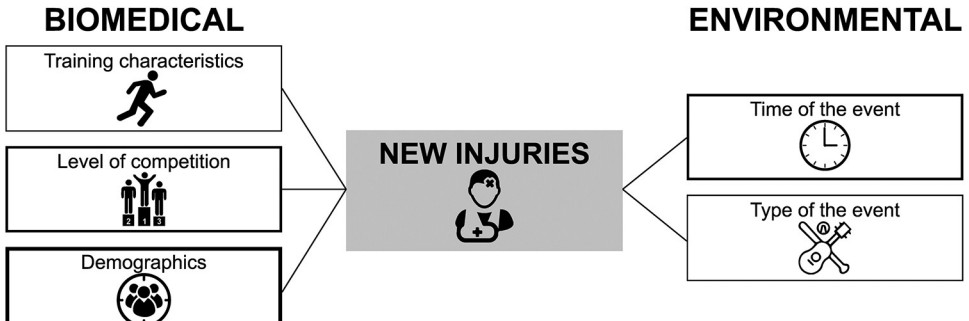

**Fig 4. Biomedical and environmental variables from multivariable regression analyses predicting new injuries.**
The thickness of the box represents the number of multivariable models including the following predictors: training characteristics (n = 1, type of training, training frequency, type of terrain); level of competition (n = 2, running pace, running experience); demographics (n = 3, age, gender, BMI); time of the event (n = 2, season); type of the event (n = 1, sport events, carnival/fetes, music concerts).

organized during both day and night (compared to day or night only) were predictive for TTHR [26]. Transport rates were highest with alcohol/drug intoxicated patients (p<0.001) and lowest with traumatic injuries (p = 0.004). Detailed information about the effect sizes of these predictors derived from multivariable models can be found in S2 Table.

### Factors that predict the incidence of new sport injuries

Three multivariable regression models to predict the incidence of new sport injuries were developed: 2 models with data of running manifestations [20, 23] and 1 model with data of a mixture of sporting manifestations, fetes/carnivals, spectator sport manifestations, concerts/ raves, and ceremonial manifestations [25]. None of the included studies reported the full equation of the multivariable model to predict the incidence of new sport injuries. The following predictor variables were included in these models: demographic information (in 3 models), type of the manifestation (in 1 model), time of the manifestation (in 2 models), level of competition (in 2 models) and training characteristics (in 1 model). Fig 4 shows the environmental and biomedical predictors for the incidence of new sport injuries derived from multivariable regression models.

The following environmental factors remained statistically significant in a multivariable model to predict new injuries: sporting manifestations and colder environmental conditions (expressed by the year of the manifestation or by season (i.e. winter versus spring)) resulted in a higher incidence of new injuries. Other types of manifestations (i.e. carnival, fete or rave concerts) or other seasons (i.e. summer vs spring; autumn versus spring) were not associated with new injuries [25].

Significant biomedical factors to predict new injuries included demographics (age and gender), level of competition and training characteristics. Older female runners during the 2 Oceans half-marathon in Cape Town (South Africa) had more incidence of medical complications (i.e. general or postural hypotension) compared to male runners and to younger female runners [20]. However, in a model with data from a mixture of sporting and non-sporting manifestations, the incidence of injuries was significantly higher in men [25]. During a (half-) marathon running race, a lower level of competition (expressed by a slower running pace (>7 minutes per km) or <5 years of running experience) and the frequency of interval training (i.e. sometimes versus always) were predictive for the incidence of new injuries. Two models

to predict injuries during half marathon races found that specific information regarding demographics (gender, BMI), level of competition (running experience, running pace category) or training characteristics (training frequency, type of terrain) did not contribute to the prediction of injury incidence. Detailed information about the effect sizes of these multivariable predictors can be found in S3 Table.

## Predictive performance of the models

Four studies reported the $R^2$ of their model to predict PPR or TTHR. The predictive performance of the PPR models ranged from very low ($R^2$ of 0.04 [16]) to (very) strong ($R^2$ values of 0.64 [15] and 0.66 [19]). The predictive accuracy of the linear TTHR model by Arbon et al was moderate ($R^2 = 0.34$) [26]. The non-linear models of Arbon et al. accurately predicted PP and TTH, as indicated by the low median error of 16 presentations per event and 1 transportation per event, respectively.

Three studies externally applied prediction models for mass gatherings by comparing the actual number of patient presentations or transports at 3 outdoor electronic dance music manifestations in the USA [28], a US spectator sport manifestations (i.e. automobile race, Baltimore Grand Prix) [29] and a city festival (i.e. Royal Air Show, Adelaide, Australia) [30] with the predicted number by the model developed by Arbon et al. [15], by Hartman et al. [32] and/or the retrospective (historical) analysis undertaken by Zeitz et al. [33]. The following predictor variables were included: weather conditions (in 3 models), crowd size (in 3 models), type of the manifestation (in 3 models), time of the manifestation (in 3 models), venue accommodation (in 3 models), presence of alcohol (in 1 model), demographic information (in 1 model).

The actual number of patient presentations and transfers to hospital in two US studies at urban auto-racing events and outdoor electronic dance music manifestations were underestimated by the Arbon/Hartman model (67–81% underestimation). The Arbon model and the Zeitz review overpredicted the actual number of casualties during the Royal Air Show in Adelaide (Australia) (22% and 10%, respectively). In this study, the actual number of daily ambulance transfers was underpredicted by 43% (Arbon model) and 53% (Zeitz review).

## GRADE assessment

Although all included studies were observational, the initial certainty level was set at 'high' because the association between predictors and outcomes was irrespective of any causal connection. The overall certainty level (for all outcomes: PPR, TTHR, injury status) was downgraded with one level (from 'high' to 'moderate') due to risk of bias since overall risk of bias was considered as 'high' in all studies (S3 and S4 Figs). Overall concerns for applicability were present in 12 studies (75%), mainly because of the limited generalizability of the study participants (concerns for applicability in 9 studies (56%)) and the outcomes assessed (concerns for applicability in 5 studies (31%)) (S5 and S6 Figs). Therefore, the certainty level was further downgraded with one level due to indirectness (from 'moderate' to 'low'). No reason was present for upgrading or further downgrading the certainty level due to imprecise or inconsistent results or publication bias.

Altogether, the final certainty in the effect estimates for the multivariable models predicting PPR, TTHR or injury status was considered as 'low'. This implies that our confidence in the effect estimates is limited and that further research is very likely to have an important impact on our confidence and is likely to change the estimate.

## Discussion

This systematic review included 16 studies that developed and/or externally applied a multi-variable regression model to predict medical usage rates at mass gatherings. We identified a set of biomedical (i.e. age, gender, level of competition, training characteristics and type of injury) and environmental predictors (i.e. crowd size, accommodation, weather, free water availability, time of the manifestation and type of the manifestation) for PPR, TTHR and injury status. No evidence for psychosocial predictors was found. The overall certainty in the effect estimates is low due to risk of bias of the studies and limited generalizability (indirectness). Evidence from the studies that applied observations from few mass gatherings to another prediction model indicated that medical usage rates are consistently over/underestimated. Therefore, the development and validation of context-specific prediction models is recommended.

To the best of our knowledge, this is the first review that systematically screened, analyzed and critically appraised studies that developed and/or validated a statistical model to predict medical usage rates at mass gatherings. Until today, numerous descriptive papers and narrative reviews on this topic have been published. For example, Nieto and Ramos found 96 articles, published between 2000 and 2015 in the Scopus database, on the type of manifestations (main type of manifestations: sports (46%), music (25%) or religious/social content (23%)) and topics covered in the mass gathering literature (main topics: health care, PPRs and/or TTHRs, respiratory pathogens, surveillance and the global spread of diseases) [34]. Moore et al. concluded that the most important predictive factors to influence medical usage rates at large manifestations were the weather, alcohol and drug use and type of manifestation [35]. Baird et al. searched 4 biomedical databases and retained 8 studies suggesting a positive relationship between temperature/humidity and PPR [36]. Our review serves as a quantitative basis to predict medical usage rates at mass gatherings by identifying those variables that were included in multivariable prediction models.

The major strength of our systematic review is the use of a rigorous methodology including sensitive search strategies in six databases, comprehensive selection criteria (no restriction to population (types of mass gatherings) or outcomes (medical usage rates)) resulting in scientific evidence, judged and critically appraised by two reviewers independently. We restricted our selection of included studies to multivariable regression models and excluded studies that only used univariate regression analyses. Advantages of multivariable analysis include the ability to represent a more realistic picture than looking at a single variable. Indeed, apparent univariate associations may in reality be explained or confounded by a non-measured predictor variable. The risk of overlooking confounding or real predictor variables decreases by including more potential predictor variables in the model. Further, multivariable techniques can control association between variables by using cross tabulation, partial correlation and multiple regressions, and introduce other variables to determine the links between the independent and dependent variables or to specify the conditions under which the association takes place. This provides a more powerful test of significance compared to univariate techniques [37]. Although some scientists have questioned the concept of statistical significance [38, 39], the statistically significant predictors from these multivariable regression models apply as the best available scientific basis for which predictors are associated with increased medical usage rates.

There are three limitations concerning the critical appraisal of the included studies design, the lack of standardized data collection and analysis, and the limited generalizability of the results. Firstly, we critically appraised the included studies by using the PROBAST checklist items [10] and the GRADE approach for the case where no single estimate of effect is present [13]. Since the GRADE working group has not yet provided specific recommendations on how to rate the certainty of effect estimates of prediction modelling studies, future formal

guidance is needed. Secondly, the methodology of data collection (both predictors and medical usage rates) and statistical analysis (i.e. different types of logistic, linear and non-linear regression analysis) varied substantially among the included studies. Hence, we were not able to conduct a meta-analysis. Although there is agreement on some broad concepts underlying mass-gathering health amongst an international group of mass gathering experts [40], more future scientific effort is needed to standardize data collection and statistical analysis when developing and/or validating a prediction model. Thirdly, most of the included prediction models were developed or validated in the USA or Australia. Since the interaction between the different biomedical, environmental, psychosocial factors and medical usage rates is complex, no extrapolation of these models to other contexts (e.g. other countries/continents, other type of manifestations, etc) can be performed. For example, climatological differences (temperature, humidity, precipitation, cloudiness, brightness, visibility, wind, and atmospheric pressure), the mixture of type of manifestations included in the prediction models (i.e. sport (spectator) manifestations, indoor/outdoor music concerts, carnivals, public exhibitions, etc.), but also difference in public health systems across countries (leading to different emergency care services delivery policies) hinders extrapolation. This limited generalizability was also confirmed by the 3 studies that applied observations from few mass gatherings to the prediction models of Arbon or Zeitz, showing significant under/overestimation of the medical usage rates when using an existing prediction model [28–30]. Future development of prediction models should therefore be validated both internally and externally, preferably against big data sets of various types of mass gatherings.

This systematic review scientifically underpinned Arbon's conceptual model with a list of statistically significant biomedical (i.e. age, gender, level of competition, training characteristics and type of injury) and environmental predictors (i.e. crowd size, accommodation, weather, free water availability, time of the manifestations and type of the manifestations) for PPR, TTHR and injury status. The $R^2$ (i.e. a statistical measure that represents the proportion of variance for medical usage rates that is explained by the biomedical/environmental predictors) of the multivariable regression models ranged from 4% to 66%. This implies that a (large) part (34–96%) of the variation in medical usage rates is as yet unexplained and dependent on unidentified factors. An important potential predictor (which is difficult to measure quantitatively) might be the characteristics of the first aid delivery services such as the amount and size of first aid posts (i.e. more posts will result in increased medical usage rates and smaller posts might result in a higher transfer to hospital rate) and the level of mobility of the first aid providers (i.e. more mobile teams will generate higher medical usage rates).

The current list of predictors are of clinical relevance for first aid or emergency services, experts and researchers involved in mass gathering. These predictors should be consistently measured in a standardized way to develop and/or validate future prediction models, in order to allow more cost-effective pre-event planning and resource provision. Another remaining question that needs to be answered in future research is how PPR and TTHR evolve over the time span of the mass gathering, in order to generate the most efficient use of (first aid) material and people (nurses, first aid providers, doctors, etc.). Since planning and preparing public health systems and services for managing a mass gathering is a complex procedure and requires a multidisciplinary approach, interdisciplinary research and international collaboration is of paramount importance to execute this future research agenda successfully [41].

## Conclusion

This systematic review identified multivariable models that predict medical usage rates at mass gatherings. Different biomedical (i.e. age, gender, level of competition, training characteristics

and type of injury) and environmental (i.e. crowd size, accommodation, weather, free water availability, time and type of the manifestation) predictors were associated with medical usage rates. Since the overall quality of the evidence is considered as low and no generic predictive model is available to date, proper development and validation of a context-specific model is recommended. Future international initiatives to standardize the collection and analysis of mass gathering health data are needed to enable the opportunity to conduct meta-analyses, to compare models across societies and modelling of various scenarios to inform health services. This will finally result in more cost-effective pre-hospital care at mass gatherings.

## Supporting information

**S1 Fig. Review authors' judgements (for each included study) on the 20 signalling questions of the 4 PROBAST domains (participants–predictors–outcome–analysis).** ⊕ Low risk of bias (answers 'yes' or 'probably yes' to signalling questions), ⊖ high risk of bias (answers 'no' or 'probably no' to signalling questions), ⍰ unclear (answer 'no information' to signalling questions). *Studies that applied observations from few mass gatherings to another prediction model (FitzGibbon 2017, Nable 2014, Zeitz 2005): items not applicable.
(TIF)

**S2 Fig. Review authors' judgements on the the 20 signalling questions of the 4 PROBAST domains (participants–predictors–outcome–analysis), presented as percentages across all included studies.**
(TIF)

**S3 Fig. Risk of bias summary: review authors' judgements on the 4 PROBAST risk of bias domains for each included study.** ⊕ Low risk of bias, ⊖ high risk of bias, ⍰ unclear.
(TIF)

**S4 Fig. Risk of bias graph: Review authors' judgements on the 4 PROBAST risk of bias domains presented as percentages across all included studies.**
(TIF)

**S5 Fig. Review authors' judgements on the applicability concerns for each included study.** ⊕ Low risk of bias, ⊖ high risk of bias, ⍰ unclear.
(TIF)

**S6 Fig. Review authors' judgements on the applicability concerns presented as percentages across all included studies.**
(TIF)

**S1 Table. Prediction model development studies for Patient Presentation Rate (PPR): Synthesis of findings of included studies.** r: correlation coefficient; RR: risk ratio; MW U: Mann-Whitney U. £ No raw data/SD's available (or specify), effect size and CI cannot be calculated; ¥ Imprecision (large variability of results); † Imprecision (lack of data).
(DOCX)

**S2 Table. Prediction model development studies for Transfer To Hospital Rate (TTHR): Synthesis of findings of included studies.** £ No raw data/SD's available (or specify), effect size and CI cannot be calculated; † Imprecision (lack of data).
(DOCX)

**S3 Table. Prediction model development studies for medical complications and injuries: Synthesis of findings of included studies.** OR: odds ratio; CI: Confidence Interval; BMI: Body Mass Index; £ No raw data/SD's available (or specify), effect size and CI cannot be

calculated; ¥ Imprecision (large variability of results); † Imprecision (lack of data).
(DOCX)

**S1 File. PRISMA checklist.**
(DOCX)

**S2 File. Detailed information on the search strategies in the different databases.**
(DOCX)

## Acknowledgments

Evi Verbecque is acknowledged for her help in developing the search strategies.

## Author Contributions

**Conceptualization:** Emmy De Buck, Philippe Vandekerckhove.

**Formal analysis:** Hans Van Remoortel, Hans Scheers.

**Funding acquisition:** Philippe Vandekerckhove.

**Investigation:** Hans Van Remoortel, Hans Scheers.

**Methodology:** Hans Van Remoortel, Hans Scheers.

**Project administration:** Hans Van Remoortel.

**Supervision:** Hans Van Remoortel, Emmy De Buck, Philippe Vandekerckhove.

**Validation:** Emmy De Buck, Winne Haenen, Philippe Vandekerckhove.

**Visualization:** Hans Van Remoortel, Hans Scheers.

**Writing – original draft:** Hans Van Remoortel.

**Writing – review & editing:** Hans Scheers, Emmy De Buck, Winne Haenen, Philippe
Vandekerckhove.

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
