## [Decision Letter · Decision Letter 0]

4 Dec 2019

PONE-D-19-28124

Prediction modelling studies for medical usage rates in mass gatherings: a systematic review

PLOS ONE

Dear Dr. Van Remoortel,

Thank you for submitting your manuscript to PLOS ONE. After careful consideration, we feel that it has merit but does not fully meet PLOS ONE’s publication criteria as it currently stands. Therefore, we invite you to submit a revised version of the manuscript that addresses the points raised during the review process.

We would appreciate receiving your revised manuscript by Jan 18 2020 11:59PM. To enhance the reproducibility of your results, we recommend that if applicable you deposit your laboratory protocols in protocols.io, where a protocol can be assigned its own identifier (DOI) such that it can be cited independently in the future. For instructions see: http://journals.plos.org/plosone/s/submission-guidelines#loc-laboratory-protocols

We look forward to receiving your revised manuscript.

Kind regards,

Ram Chandra Bajpai, Ph.D.

Academic Editor

PLOS ONE

Journal Requirements:

1. PLOS requires an ORCID iD for the corresponding author in Editorial Manager on papers submitted after December 6th, 2016. Please ensure that you have an ORCID iD and that it is validated in Editorial Manager. To do this, go to ‘Update my Information’ (in the upper left-hand corner of the main menu), and click on the Fetch/Validate link next to the ORCID field. This will take you to the ORCID site and allow you to create a new iD or authenticate a pre-existing iD in Editorial Manager. Please see the following video for instructions on linking an ORCID iD to your Editorial Manager account: https://www.youtube.com/watch?v=_xcclfuvtxQ

Reviewers' comments:

Reviewer's Responses to Questions

**Comments to the Author**

1. Is the manuscript technically sound, and do the data support the conclusions?

Reviewer #1: Partly

Reviewer #2: Yes

Reviewer #3: Yes

2. Has the statistical analysis been performed appropriately and rigorously? 

Reviewer #1: N/A

Reviewer #2: Yes

Reviewer #3: Yes

3. Have the authors made all data underlying the findings in their manuscript fully available?

Reviewer #1: Yes

Reviewer #2: Yes

Reviewer #3: Yes

4. Is the manuscript presented in an intelligible fashion and written in standard English?

Reviewer #1: No

Reviewer #2: Yes

Reviewer #3: Yes

5. Review Comments to the Author

Reviewer #1: The authors present a systematic review of prediction models for outcomes such as patient presentations and transfers to hospital at mass gatherings. I have the following comments and suggestions for the authors:

• The aim of the review is rather broad and could have been refined a little further. At first I thought it was about evaluating the prediction models, however, the results focus on predictors that were found to be statistically significant in the models, suggesting interest in association rather than the model’s ability to predict.

o If interested in predictor associations, why not consider all evidence for predictors rather than limiting to multivariable models? Can the authors at least clarify if they included predictor finding studies if they were multivariable i.e. if they included adjustment factors, even if the models weren’t intended to be used for prediction?

o Although univariable models were supposedly excluded in the review, some remain. The model by Kman is a univariable model as it only includes temperature. Crowd size is not a predictor as such. The model has just been rearranged so rather than modelling the rate, what would be the denominator (crowd size) is on the other side of the equation. The model(s) by Grange were univariable too.

o There seems to be a reliance on p-values without consideration of sample size. If studies were large enough, p-values will be significant for small effects and potentially important effects could be overlooked due to small sample size. If the focus is on individual predictors, then perhaps meta-analysis of (adjusted) effects (where reported) could have been performed to arrive at overall conclusions for these factors rather than discounting them due to p-values in individual studies, or at least considering all reported associations rather than significant ones.

o The quantities of interest are not clear in the methods. If interested in prediction models, I would have expected to see measures of predictive performance (more than just R2) such as measures of calibration. Instead a range of quantities were reported including median error which isn’t clear what that means.

• If the focus is on predictor effects, why were external validation studies included? If this is a secondary aim of the review, then perhaps it could be separated. For example, table 1 includes validation studies with lists of candidate predictors, yet a validation study would assess performance of the model and the predictors included are fixed. Also, included ‘validation studies’ in this review includes models that were applied for a single gathering, which is just a prediction (which is then compared to the observed rates) rather than a proper validation study. This should at least be discussed.

Other minor comments:

• Might be better to refer to ‘events’ as mass gatherings as ‘event’ may be confused with outcome.

• Rather than ‘multivariate’, I think the authors mean ‘multivariable’ i.e. multiple variables in a model rather than modelling multiple outcomes. This is rather confusing when some studies developed models (in particular page 8, lines 15-16).

• Page 8, results: In addition to total mass gatherings, would be helpful to give the range and median.

• Table 1: don’t agree that studies are cross-sectional as gatherings occurred at different times. Several are databases (sometimes retrospective) of gatherings.

• Table 1 or S1-3: Would be useful to know which type of model was used for each outcome e.g. linear, Poisson etc.

• Please correct language throughout. A few examples include, page 7, line 14: ‘model against which the actual data were validated’, page 15 line 8: multivariate predictors,

• Not clear if most models were reported in a way that could be used for prediction e.g reporting the full model equation including the intercept or just predictor effects.

Reviewer #2: Thank you so much for giving me the opportunity to review this important manuscript.

This work provides information to aid effective planning of mass gatherings, which are on the increase in every society. The authors pointing to the fact that the United Nations, through the World Health Organization, is paying serious attention to mass gatherings, with the recommendation of mass gathering medicine for all the WHO member states. For the relevance of the topic, I score this manuscript high and suggest that it should be accepted for publication, subject to addressing the obvious gaps I highlighted in each of the sections below (introduction, results, discussion and conclusion):

Introduction Section

The justification for the study is weak. The main justification that the authors provide for this study is contained in the two sentences below:

“Since mass gatherings attended by large crowds have become a more frequent feature of society, mass gathering medicine was highlighted as a new discipline at the World Health Assembly of Ministers of Health in Geneva in May 2014. As a consequence, the amount of international initiatives and meetings on mass gathering medicine has increased over the past decade as has the number of experts and the amount of publications on pre-event planning and surveillance for mass gathering”.

However, the frequency of mass gathering, the recommendation of mass gathering medicine for the WHO member states by the WHO and the increase in international initiatives and meetings on mass gathering medicine, are not a strong public health/clinical problem to justify a study. It is important to clearly state the specific clinical, public health and economic risks associated with mass gatherings to justify this study. For instance, do mass gatherings increase the risks of diseases and mortality to the participants and the larger society? What are the economic and social problems associated with mass gatherings that may dove tail into major health problems? The authors have mentioned “Patient presentation rate (PPR), Transfer to Hospital Rate (TTHR) and new injuries” as the major health problems but these alone cannot attract the attention of policy makers to invest in interventions to minimize the risks of mass gatherings. In the developing countries, there are far more important public health and developmental problems than PPR, TTHR and new injuries, to attract the attention of government. Therefore the authors must show what the society is suffering on account of mass gathering to strongly justify this study.

The authors also need to show how poor planning of events increase the risks associated with mass gathering. This is so because the findings of the study are intended to promote effective pre-event planning. If policy makers cannot see the risks of poor event planning, why should they care to make policies that would encourage effective planning?

Results Section

The result section has six sections: (1) Study selection, (2) factors that predict patients’ presentation (rate) (3) Factors that predict transfer to hospital (rate) (4) factors that predict the incidence of new sport injuries (5) external model validation studies and (6) Graded assessment. My comments on each of the sections are provided below:

(1) Study selection,

This section is too detailed and elaborate. Some of the information provided in this section should be taken to the methods section. For instance, the following sentences can go to the methods:

“Fig 1 represents the study selection process used” – This sentence has nothing to do with results

“A mix of different types of mass events was included such as sports (spectator) events (e.g. soccer games, auto races, (half-) marathon), music concerts (indoor/outdoor), fete/carnivals, public exhibitions and ceremonial events”. This sentence has nothing to do with results. It’s a method statement.

“Data were collected in 8 studies between 2005-2015, in 7 studies between 1995-2005, and in 2 studies between 1980-1995”.

The above sentences are not part of the results and therefore should not be included in the results.

We can summarize the entire section by providing only significant information that tally with the aim of the study. Here is my recommendation below:

“We included 17 cross sectional studies and more than 1,700 mass events (more than 48 million people attending these vents). Majority of the studies (n=13, 76%) were conducted in the USA (n=9, 52.94%) and Australia (n=4, 23.52%); with a few studies from Japan (n=1, 5.88%), Singapore (n=1, 5.88%), South Africa (n=1, 5.88%) and The Netherlands (n=1, 5.88%). Most of the studies [n= 15, 88.23%) measured influx at first aid posts as an outcome, while nearly half of the studies (n=7, 41.17%) focused on transfer to the hospital, with a few studies (n=3, 17.64%) assessing the incidence of new (non-)medical injuries/complications as the outcome. Almost all the studies investigated whether at least one of the following environmental candidate predictors were associated with medical usage (rates): Thirteen studies (76.47%) assessed weather; 12 (70.58%) studies assessed crowd size; etc”

(2) Factors that predict patients’ presentation (rate)

Again, there are several sentences in this section that should be moved to the discussion section. Any sentence that clarifies or justifies an observation should go to the discussion, e. g., this sentence:

“In a recent study by Arbon et al., temperature (i.e. <23.5°C vs ≥23.5°C and <25.5°C vs ≥25.5°C) was included in a non-linear regression tree model with a lower total number of patient presentations in case of higher temperatures”. This sentence is not a result but supporting information. Please note that in the result section, the guidelines requires only the findings to be presented without justifying or supporting them with references. We can do this in the discussion section to illustrate the consistency of our findings with the findings of other researchers.

In order to portray the significance of the factors and their contributions, I suggest that percentages (%) should be used instead of just mentioning the factors and the number of studies that reported them. If the data will allow, the P-values should also be reported in all the predictable factors. Importantly, note that this sections deals with only the predictors and not the studies.

(3) Factors that predict transfer to hospital (rate)

The comments made in (2) above applies here also.

(4) Factors that predict the incidence of new sport injuries

The comments made in (2) above applies here also.

Discussion

The first paragraph of the discussion is a repetition of information that is well documented in previous sections. Perhaps, starting the discussion with the major findings of this study will be more exciting as an introduction to the discussion than repeating the work that was done.

The findings of the study should also be discussed in line with the findings of other authors, since it was reported in the introduction that there are has been an “increased over the past decade in the number of experts and the amount of publications on pre-event planning and surveillance for mass gathering”.

Conclusion

I suggest that the conclusion should be revised to show that the two major aims of conducting the study have been achieved. The implications of the findings should also be stated.

Reviewer #3: Hans Van Remoortel et al have done a systematic review of prediction models for medical usage rates in mass gatherings. It is a well reported and conducted study and authors have appropriately used the PRISMA checklist. I had a few queries, however, and have the following comments and suggestions to further strengthen the paper:

1. Multivariate refers to analyses used in longitudinal studies where outcomes are collected over multiple time points. I do not think this is what the authors mean when they use this term in their paper. Prediction models are often multivariable, where multiple predictors are used in the right-hand side of the model, not multivariate. Please could authors clarify and make necessary changes throughout the paper, including the supplementary information.

2. Authors use the CHARMS checklist to assess the risk of bias of the included risk prediction models, which is designed for the critical appraisal and data extraction for systematic reviews of prediction models and not necessarily/explicitly the risk of bias of prediction models. Please discuss how this tool was used to assess the risk of bias of the included prediction modelling studies. PROBAST: A Tool to Assess the Risk of Bias and Applicability of Prediction Model Studies was published earlier this year; did authors consider assessing the risk of bias of the prediction models using PROBAST?

3. Page 2, line 25 (abstract): what is meant by ‘(low certainty evidence)’?

4. Page 2, line 25 (abstract): last sentence reads more like a conclusion than a result. Can authors clarify this?

5. Page 2, line 29-30: I don’t think this conclusion is supported by what written in the results of the abstract.

6. The selection of papers is slightly confusing, and the checking of conflicts seems to occur late in the process. Reviewers independently review the publications and only resolved disagreements at the final stage. This is after title and abstract, and full text screening, and I wonder why this process was chosen and what impact this might have had on the results? Did authors consider checking disagreements earlier, after title and abstract screening? How did they ensure eligible papers were not discarded, as results are only checked at the final stage after reviewers may have excluded an eligible paper?

7. Page 7, line 5: please amend ellipsis in ‘(sport, music, public exhibitions, …)’.

8. Page 8, line 4: rethink/omit use of word ‘scrutinise’.

9. Page 8, line 5: remove ‘eventually’.

10. Page 8, lines 8-10: numbers and percentages to support statement needed.

11. Page 8, lines 12-13: please out in chronological order for easier readability.

12. Table 1: a column for regression type used would be useful.

13. Table 1: could the sample size be more clearly written – total sample and number of events?

14. Table 1: could total number of predictors included in the final model also be included?

15. References in the results are confusing – could the reference numbers be added to Table 1 so it is easier to link included papers to reference numbers.

6. PLOS authors have the option to publish the peer review history of their article (what does this mean?). If published, this will include your full peer review and any attached files.

Reviewer #1: No

Reviewer #2: No

Reviewer #3: Yes: Dr Paula Dhiman

---

## [Author Response · Author response to Decision Letter 0]

9 Mar 2020

Reviewer #1: The authors present a systematic review of prediction models for outcomes such as patient presentations and transfers to hospital at mass gatherings. I have the following comments and suggestions for the authors:

∙ The aim of the review is rather broad and could have been refined a little further. At first I thought it was about evaluating the prediction models, however, the results focus on predictors that were found to be statistically significant in the models, suggesting interest in association rather than the model’s ability to predict.

Comment 1

o If interested in predictor associations, why not consider all evidence for predictors rather than limiting to multivariable models? 

Answer: 

We used evidence from multivariable models because these type of models allow to look at relationships between variables in an overarching way and to quantify the relationship between variables. These models can control association between variables (e.g. by using cross tabulation, partial correlation and multiple regressions) and introduce other variables to determine the links between the independent and dependent variables. The major advantages of using multivariate analysis include an ability to have a more realistic picture, rather than looking at a single variable. Further, multivariate techniques provide a powerful test of significance compared to univariate techniques. We clarified our rationale to use evidence from multivariate models in the methods section of our manuscript (page 6, line 29 & page 7 lines 1-4).

Can the authors at least clarify if they included predictor finding studies if they were multivariable i.e. if they included adjustment factors, even if the models weren’t intended to be used for prediction?

Answer: 

Most studies included in our review claimed that they were intended to be used for prediction. However, some of these described associations without actually predicting PPR or TTHR (e.g. Grange 1999, Locoh-Donou 2016). Other studies (e.g. van Poppel 2016) aimed to identify risk factors for being a casualty at mass gatherings, rather than to predict PPR. All studies showing multivariable associations between risk factors and PPR were included in our review, regardless of whether the models presented could readily predict PPR or not, because the associations shown provide useful information for pre-event planning and resource provision at mass gatherings by organizations like ours (Red Cross).

Comment 2

o Although univariable models were supposedly excluded in the review, some remain. The model by Kman is a univariable model as it only includes temperature. Crowd size is not a predictor as such. The model has just been rearranged so rather than modelling the rate, what would be the denominator (crowd size) is on the other side of the equation. The model(s) by Grange were univariable too.

Answer:

Indeed, the model by Grange et al. (1999) ended up being univariable. They evaluated four possible predictors (attendance, temperature, indoor/outdoor, music category), but since only music category was significant in a univariable model, no multivariable model was built. So our criterion for including a study was the intention to evaluate more than one predictor variable in a multivariable model, regardless of how many predictor variables remained in the final model. We clarified this in the Methods section (page 7, lines 2-4) of the revised manuscript. 

Concerning the study by Kman et al. (2007), we agree with the reviewer that the model is actually univariable. As temperature was the only predictor variable right from the start, this study does not fulfill our inclusion criteria and we removed it from our revised review.

Comment 3

o There seems to be a reliance on p-values without consideration of sample size. If studies were large enough, p-values will be significant for small effects and potentially important effects could be overlooked due to small sample size. 

Answer:

We fully agree with the reviewer that sample size is an important factor when judging confidence in the effect estimates observed (conclusions). Sample size was considered when assessing the risk of bias of the studies, using PROBAST (cfr. infra). In prediction model studies, overall sample size matters, but the number of participants with the outcome is even more important. Sample size considerations for model development studies have historically based on the number of events per variable (EPV) or per predictor. In general, studies with EPVs lower than 10 are likely to have overfitting, whereas those with EPVs higher than 20 are less likely to have overfitting (Moons et al., Annals of Internal Medicine, 2019, pmid: 30596876). We used the PROBAST tool to critically appraise the method of analysis used in the individual studies, including the sample size used (see Figure S1 – signalling question ‘ANALYSIS – Were there a reasonable number of participants with the outcome?’). We finally downgraded the level of evidence due to overall high risk of bias (also risk of bias related to ‘analysis’ was considered as ‘high’, see results section page 22, lines 11-18)

Comment 5

If the focus is on individual predictors, then perhaps meta-analysis of (adjusted) effects (where reported) could have been performed to arrive at overall conclusions for these factors rather than discounting them due to p-values in individual studies, or at least considering all reported associations rather than significant ones.

Answer: 

We were unable to perform any meta-analysis because of the large variation in predictor variables, outcome definition (PPR, total presentations, injury status) and modelling techniques (linear, logistic or Poisson regression, or simple correlations). Moreover, some studies publish parameter estimates without standard errors. Therefore, not a single set of estimates, sufficiently similar to each other to be pooled in a meta-analysis, could be formed.

Comment 6

The quantities of interest are not clear in the methods. If interested in prediction models, I would have expected to see measures of predictive performance (more than just R2) such as measures of calibration. Instead a range of quantities were reported including median error which isn’t clear what that means.

Answer: 

The main quantity of interest is the prediction model itself (if provided), as shown in Table S1. In the revised manuscript, measures of predictive performance are evaluated within the PROBAST tool (see Figure S1 – signalling question ‘ANALYSIS – Were relevant model performance measures evaluated appropriately?’). The evaluation shows that only one study reported a measure of predictive performance: the median error by Arbon 2018. This median error is the median difference between observed and predicted patient presentations for all included manifestations. Because median error is a measure of predictive performance and not an effect size itself, we removed it from the “effect size” column Table S1. Thus, the term median error is not mentioned anymore in the text or tables and needs no further explanation.

Comment 7

If the focus is on predictor effects, why were external validation studies included? If this is a secondary aim of the review, then perhaps it could be separated. For example, table 1 includes validation studies with lists of candidate predictors, yet a validation study would assess performance of the model and the predictors included are fixed. 

Answer: 

We were both interest in predictors from new prediction development models/studies as well as the internal/external validation of these predictors. Therefore, our general study aim was “to identify studies that developed or validated a multivariate statistical model to predict medical usage rates at mass gatherings” (page 5, lines 7-9). We agree with the reviewer that results from development and validation studies should be separated for clarification purposes. Therefore, we sorted the included studies in table 1 per study design: prediction model development studies without external validation vs prediction model development studies with internal validation vs external model validation studies. In the results section of the manuscript, the results from the prediction model studies were separated from the external validation studies. 

Comment 8

Also, included ‘validation studies’ in this review includes models that were applied for a single gathering, which is just a prediction (which is then compared to the observed rates) rather than a proper validation study. This should at least be discussed.

Answer: 

Validation studies were defined according to Moons et al (see reference 9) as “studies that aimed to assess and compare the predictive performance of an existing prediction model (i.e. models by Arbon/Zeitz in the 3 included validation studies) using new participant data that were not used to develop the prediction model (i.e. data from 3 electronic dance music festivals (Fitzgibbon 2017), Baltimore Grand Prix (Nable 2014) and Royal Adelaide Show (Zeitz 2005)) and that possibly adjust or update the model in case of poor performance based on the validation data (i.e. models in these 3 studies were not updated).”

Other minor comments:

Comment 9

Might be better to refer to ‘events’ as mass gatherings as ‘event’ may be confused with outcome.

Answer: 

We agree with the reviewer and changed ‘events’ to ‘manifestations’ throughout the manuscript.

Comment 10

Rather than ‘multivariate’, I think the authors mean ‘multivariable’ i.e. multiple variables in a model rather than modelling multiple outcomes. This is rather confusing when some studies developed models (in particular page 8, lines 15-16).

Answer: 

We agree with the reviewer and changed this accordingly.

Comment 11

Page 8, results: In addition to total mass gatherings, would be helpful to give the range and median.

Answer: 

We agree with the reviewer and added this information in the manuscript (page 10, lines 7-8).

Comment 12

Table 1: don’t agree that studies are cross-sectional as gatherings occurred at different times. Several are databases (sometimes retrospective) of gatherings.

Answer: 

We agree with the reviewer and removed the term ‘cross-sectional’

Comment 13

Table 1 or S1-3: Would be useful to know which type of model was used for each outcome e.g. linear, Poisson etc.

Answer: 

We agree with the reviewer and added this information in an extra column in table 1.

Comment 14

Please correct language throughout. A few examples include, page 7, line 14: ‘model against which the actual data were validated’, page 15 line 8: multivariate predictors,

Answer: 

We agree with the reviewer and changed this accordingly

Comment 15

Not clear if most models were reported in a way that could be used for prediction e.g reporting the full model equation including the intercept or just predictor effects.

Answer:

We agree with the reviewer and added a column in tables S1-S3 to indicate whether a full model equation was available (or not). Summarized information was added in the results section of the manuscript (page 19, lines 13-14; page 20, line 11; page 21, lines 13-15). 

Reviewer #2: Thank you so much for giving me the opportunity to review this important manuscript.

This work provides information to aid effective planning of mass gatherings, which are on the increase in every society. The authors pointing to the fact that the United Nations, through the World Health Organization, is paying serious attention to mass gatherings, with the recommendation of mass gathering medicine for all the WHO member states. For the relevance of the topic, I score this manuscript high and suggest that it should be accepted for publication, subject to addressing the obvious gaps I highlighted in each of the sections below (introduction, results, discussion and conclusion):

Introduction Section

Comment 1

The justification for the study is weak. The main justification that the authors provide for this study is contained in the two sentences below:

“Since mass gatherings attended by large crowds have become a more frequent feature of society, mass gathering medicine was highlighted as a new discipline at the World Health Assembly of Ministers of Health in Geneva in May 2014. As a consequence, the amount of international initiatives and meetings on mass gathering medicine has increased over the past decade as has the number of experts and the amount of publications on pre-event planning and surveillance for mass gathering”.

However, the frequency of mass gathering, the recommendation of mass gathering medicine for the WHO member states by the WHO and the increase in international initiatives and meetings on mass gathering medicine, are not a strong public health/clinical problem to justify a study. It is important to clearly state the specific clinical, public health and economic risks associated with mass gatherings to justify this study. For instance, do mass gatherings increase the risks of diseases and mortality to the participants and the larger society? What are the economic and social problems associated with mass gatherings that may dove tail into major health problems? The authors have mentioned “Patient presentation rate (PPR), Transfer to Hospital Rate (TTHR) and new injuries” as the major health problems but these alone cannot attract the attention of policy makers to invest in interventions to minimize the risks of mass gatherings. In the developing countries, there are far more important public health and developmental problems than PPR, TTHR and new injuries, to attract the attention of government. Therefore the authors must show what the society is suffering on account of mass gathering to strongly justify this study.

The authors also need to show how poor planning of events increase the risks associated with mass gathering. This is so because the findings of the study are intended to promote effective pre-event planning. If policy makers cannot see the risks of poor event planning, why should they care to make policies that would encourage effective planning?

Answer:

We agree with the reviewer and provided additional introductory information to justify our review as follows (pages 4, lines 10-21): “Mass gatherings are associated with increased health risks and hazards such as the transmission of communicable diseases, exacerbation of non-communicable diseases and comorbidities (e.g. diabetes, hypertension, COPD, cardiovascular events) and an impact on mental/physical health and psychosocial disorders.” (ref pmid 31106753) 

Furthermore, the mental health consequences of traumatic incidents at mass gatherings can be prolonged with stress to people, families, and communities resulting in short-term fear of death as well as general distress, anxiety, excessive alcohol consumption, and other psychiatric disorders. If mass gatherings are improperly managed, this can lead to human, material, economic or environmental losses and impacts (pmid 27062983)

Therefore, the development of (cost-)effective methods for the planning and handling of the health risks associated with mass gatherings will strengthen global health security, prevent excessive emergency health problems and associated economic loss, and mitigate potential societal disruption in host and home communities (pmid 22252148).”

Results Section

Comment 2

The result section has six sections: (1) Study selection, (2) factors that predict patients’ presentation (rate) (3) Factors that predict transfer to hospital (rate) (4) factors that predict the incidence of new sport injuries (5) external model validation studies and (6) Graded assessment. My comments on each of the sections are provided below:

(1) Study selection,

This section is too detailed and elaborate. Some of the information provided in this section should be taken to the methods section. For instance, the following sentences can go to the methods:

“Fig 1 represents the study selection process used” – This sentence has nothing to do with results

“A mix of different types of mass events was included such as sports (spectator) events (e.g. soccer games, auto races, (half-) marathon), music concerts (indoor/outdoor), fete/carnivals, public exhibitions and ceremonial events”. This sentence has nothing to do with results. It’s a method statement.

“Data were collected in 8 studies between 2005-2015, in 7 studies between 1995-2005, and in 2 studies between 1980-1995”.

The above sentences are not part of the results and therefore should not be included in the results.

We can summarize the entire section by providing only significant information that tally with the aim of the study. Here is my recommendation below:

“We included 17 cross sectional studies and more than 1,700 mass events (more than 48 million people attending these vents). Majority of the studies (n=13, 76%) were conducted in the USA (n=9, 52.94%) and Australia (n=4, 23.52%); with a few studies from Japan (n=1, 5.88%), Singapore (n=1, 5.88%), South Africa (n=1, 5.88%) and The Netherlands (n=1, 5.88%). Most of the studies [n= 15, 88.23%) measured influx at first aid posts as an outcome, while nearly half of the studies (n=7, 41.17%) focused on transfer to the hospital, with a few studies (n=3, 17.64%) assessing the incidence of new (non-)medical injuries/complications as the outcome. Almost all the studies investigated whether at least one of the following environmental candidate predictors were associated with medical usage (rates): Thirteen studies (76.47%) assessed weather; 12 (70.58%) studies assessed crowd size; etc”

Answer:

According to items 19 and 20 of the PRISMA checklist (see Appendix S1), describing details on the study selection and the characteristics of included studies is an essential part (first paragraph) of the results section, and also in other systematic reviews the results of the study selection and the characteristics of the identified studies are described as a result (and not as a method). Therefore, we would like to keep this information in the results section. 

Comment 3

(2) Factors that predict patients’ presentation (rate)

Again, there are several sentences in this section that should be moved to the discussion section. Any sentence that clarifies or justifies an observation should go to the discussion, e. g., this sentence:

“In a recent study by Arbon et al., temperature (i.e. <23.5°C vs ≥23.5°C and <25.5°C vs ≥25.5°C) was included in a non-linear regression tree model with a lower total number of patient presentations in case of higher temperatures”. This sentence is not a result but supporting information. Please note that in the result section, the guidelines requires only the findings to be presented without justifying or supporting them with references. We can do this in the discussion section to illustrate the consistency of our findings with the findings of other researchers.

(3) Factors that predict transfer to hospital (rate)

The comments made in (2) above applies here also.

(4) Factors that predict the incidence of new sport injuries

The comments made in (2) above applies here also.

Answer:

The result of a systematic review is a number of included studies answering the research/PICO question. The results described here are data from one/several of the included studies (e.g. Arbon et al 2018; this is not a supporting reference but one of the included studies from our systematic review, and therefore “a result”, we changed this accordingly in the manuscript as follows: “in one of our included studies….”). Therefore, the authors believe that this is appropriate information (results + corresponding reference) for the results section. 

Comment 4

In order to portray the significance of the factors and their contributions, I suggest that percentages (%) should be used instead of just mentioning the factors and the number of studies that reported them. If the data will allow, the P-values should also be reported in all the predictable factors. Importantly, note that this sections deals with only the predictors and not the studies.

Answer:

We agree with the reviewer and reported additional information regarding ALL predictor variables included in the multivariable models (not only the statistical significant ones) (Page 18, lines 2-6; page 19, lines 23-26; page 20, lines 24-29). Since we were not able to perform meta-analyses we were not able to report individual p-values for each predictor variable/study. We provided the available individual p-values in the supplementary tables S1-S2-S3. In the manuscript, we summarized this information by defining statistically significant as a p-value below 0.05 (cfr. methods section, page 8, line 19).

Discussion

Comment 5

The first paragraph of the discussion is a repetition of information that is well documented in previous sections. Perhaps, starting the discussion with the major findings of this study will be more exciting as an introduction to the discussion than repeating the work that was done.

Answer:

The authors believe that the first paragraph of the discussion reflects the major findings (cfr. aim at the end of the introduction) of our systematic review, namely: 1) the number of prediction development and validation studies included, 2) providing an overview of environmental and biomedical factors that were associated with medical usage rates and 3) major conclusion regarding the quality of the included studies.

Comment 6

The findings of the study should also be discussed in line with the findings of other authors, since it was reported in the introduction that there are has been an “increased over the past decade in the number of experts and the amount of publications on pre-event planning and surveillance for mass gathering”.

Answer:

As described in the second paragraph of the discussion (page 23, lines 12-14), we addressed that our manuscript is the first systematic review (according to rigorous methodological standards (cfr. Cochrane handbook) that systematically analyzed and critically appraised the available literature in this field (cfr. our PICO question). With this review, we were able to capture all relevant papers in this field, answering our PICO question. In this second paragraph, we compared our findings with other non-systematic (i.e. descriptive/narrative) reviews in this field (page 23, lines 14-24).

Comment 7

Conclusion

I suggest that the conclusion should be revised to show that the two major aims of conducting the study have been achieved. The implications of the findings should also be stated.

Answer:

We agree with the reviewer and rephrased the conclusion as follows (page 25 lines 27-30 & page 26 lines 1-6): “This systematic review identified multivariable models that predict medical usage rates at mass gatherings. Different biomedical (i.e. age, gender, level of competition, training characteristics and type of injury) and environmental (i.e. crowd size, accommodation, weather, free water availability, time of the events and type of the events) predictors were associated with medical usage rates. Since the overall quality of the evidence is low and no generic predictive model is available today, proper development and validation of a context-specific model is recommended. Future international initiatives to standardize the collection and analysis of mass gathering health data are needed to enable the possibility for meta-analyses, comparison of events across societies and modelling of various scenarios to inform health services. This will finally result in a more cost-effective pre-hospital care at mass gatherings.”

Reviewer #3: Hans Van Remoortel et al have done a systematic review of prediction models for medical usage rates in mass gatherings. It is a well reported and conducted study and authors have appropriately used the PRISMA checklist. I had a few queries, however, and have the following comments and suggestions to further strengthen the paper:

Comment 1

Multivariate refers to analyses used in longitudinal studies where outcomes are collected over multiple time points. I do not think this is what the authors mean when they use this term in their paper. Prediction models are often multivariable, where multiple predictors are used in the right-hand side of the model, not multivariate. Please could authors clarify and make necessary changes throughout the paper, including the supplementary information.

Answer:

We agree with the reviewer and changed the wording (i.e. multivariable instead of multivariate) accordingly throughout the manuscript.

Comment 2

Authors use the CHARMS checklist to assess the risk of bias of the included risk prediction models, which is designed for the critical appraisal and data extraction for systematic reviews of prediction models and not necessarily/explicitly the risk of bias of prediction models. Please discuss how this tool was used to assess the risk of bias of the included prediction modelling studies. PROBAST: A Tool to Assess the Risk of Bias and Applicability of Prediction Model Studies was published earlier this year; did authors consider assessing the risk of bias of the prediction models using PROBAST?

Answer:

We thank the reviewer for this valuable comment. Since the paper of the PROBAST tool was published after we submitted our manuscript, we were not able to use this tool initially. However, we agree with the reviewer (and double checked with the first author of the PROBAST paper, Karel Moons) that the PROBAST tool is the most recent tool to assess the risk of bias compared to the CHARMS checklist. Therefore, the 2 reviewers independently used the PROBAST tool to reassess the risk of bias (20 signalling questions on 4 domains) and checked for applicability concerns for the 16 included studies. We added or changed information in the methods and results section accordingly (page 8, lines 3-16; page 22, lines 13-28).

Comment 3

Page 2, line 25 (abstract): what is meant by ‘(low certainty evidence)’?

Answer:

The GRADE methodology was used to assess the certainty of the evidence (also known as quality of evidence or confidence in effect estimates). Low certainty of the evidence means that the likelihood is high that the true effect of a predictor variable may be substantially different from its effect estimated by the research and, hence, that knowledge of the true effect would lead to different practical decisions than the estimated effect. 

According to the GRADE methodology, judgements about the certainty of the evidence are based on factors that reduce the certainty (risk of bias, inconsistency, indirectness, imprecision, publication bias) and factors that increase the certainty, such as large magnitude of effect. For clarification purposes, we changed the information regarding the GRADE methodology in the methods section (page 7, lines 20-29 & page 8, lines 1-3). 

Comment 4

Page 2, line 25 (abstract): last sentence reads more like a conclusion than a result. Can authors clarify this?

Answer:

Our review included 14 prediction model development studies (with or without internal validation) and 3 external validation studies (without model updating), see table 1. This last sentence refers to the key results from the 3 external validation studies. We rephrased this sentence as follows: “Evidence from the external validation studies indicated that using Arbon’s or Zeitz’ model in other contexts significantly under- or overestimated medical usage rates (from 22% overestimation to 81% underestimation)”

Comment 5

Page 2, line 29-30: I don’t think this conclusion is supported by what written in the results of the abstract.

Answer:

We agree with the reviewer and changed the conclusion as follows: “This systematic review identified multivariable models that predict medical usage rates at mass gatherings. Different biomedical/environmental predictors were associated with medical usage rates. Since the overall quality of the evidence is low and no generic predictive model is available today, proper development and validation of your context-specific model is recommended.”

Comment 6

The selection of papers is slightly confusing, and the checking of conflicts seems to occur late in the process. Reviewers independently review the publications and only resolved disagreements at the final stage. This is after title and abstract, and full text screening, and I wonder why this process was chosen and what impact this might have had on the results? Did authors consider checking disagreements earlier, after title and abstract screening? How did they ensure eligible papers were not discarded, as results are only checked at the final stage after reviewers may have excluded an eligible paper?

Answer:

All abstracts/papers included after title and abstract screening were considered and discussed between the two reviewers, but not immediately after title and abstract screening but after full text assessment. This was done because information required to make a judgement on eligibility is often not available in the abstract. The independent title/abstract/full text screening by the 2 reviewers together with screening of the 20 first related citations in Pubmed for the included studies were maximal efforts to prevent missing eligible papers. We rephrased the relevant sentence in the methods as follows: “Two authors (HVR and HS) independently screened the titles and abstracts of all references yielded by the search. Subsequently, the full text of each article that potentially met the eligibility criteria was obtained, and after a full-text assessment, studies that did not meet the selection criteria were excluded. Any discrepancies between authors was resolved by consensus or by consulting a third reviewer (EDB).”

Comment 7

Page 7, line 5: please amend ellipsis in ‘(sport, music, public exhibitions, …)’.

Answer:

We changed this accordingly.

Comment 8

Page 8, line 4: rethink/omit use of word ‘scrutinise’.

Answer:

We changed ‘scrutinised’ into ‘screened’.

Comment 9

Page 8, line 5: remove ‘eventually’.

Answer:

We changed this accordingly.

Comment 10

Page 8, lines 8-10: numbers and percentages to support statement needed.

Answer:

We added this information accordingly

Comment 11

Page 8, lines 12-13: please out in chronological order for easier readability.

Answer:

We rephrased this sentence accordingly.

Comment 12

Table 1: a column for regression type used would be useful.

Answer:

We agree with the reviewer and provided this information in an additional column.

Comment 13

Table 1: could the sample size be more clearly written – total sample and number of events?

Answer:

We agree with the reviewer and clarified this explicitly in the column ‘Population – mass gathering events’

Comment 14

Table 1: could total number of predictors included in the final model also be included?

Answer:

We added this information in the column ‘candidate predictors’

Comment 15

References in the results are confusing – could the reference numbers be added to Table 1 so it is easier to link included papers to reference numbers.

Answer: 

We agree with the reviewer and added the reference in the first column.

---

## [Decision Letter · Decision Letter 1]

18 May 2020

PONE-D-19-28124R1

Prediction modelling studies for medical usage rates in mass gatherings: a systematic review

PLOS ONE

Dear Dr. Van Remoortel,

Thank you for submitting your manuscript to PLOS ONE. After careful consideration, we feel that it has merit but does not fully meet PLOS ONE’s publication criteria as it currently stands. Therefore, we invite you to submit a revised version of the manuscript that addresses the points raised during the review process.

We would appreciate receiving your revised manuscript by Jul 02 2020 11:59PM. To enhance the reproducibility of your results, we recommend that if applicable you deposit your laboratory protocols in protocols.io, where a protocol can be assigned its own identifier (DOI) such that it can be cited independently in the future. For instructions see: http://journals.plos.org/plosone/s/submission-guidelines#loc-laboratory-protocols

We look forward to receiving your revised manuscript.

Kind regards,

Tim Mathes

Academic Editor

PLOS ONE

Additional Editor Comments (if provided):

Editor:

This is an interesting manuscript. However, before beinig considered for publication some minor issues in the methods section should be clarified.

1. Please clarify how many reviewers performed the risk of bias assessment. In addition, the risk of bias assessment should be described in an own section. In general, please try to use headings as suggested in PRISMA.

2. GRADE was not developed to assess studies on risk prediction models. Therefore, the conduct of the GRADE assessment should be described in much more detail.

Reviewers' comments:

Reviewer's Responses to Questions

**Comments to the Author**

1. If the authors have adequately addressed your comments raised in a previous round of review and you feel that this manuscript is now acceptable for publication, you may indicate that here to bypass the “Comments to the Author” section, enter your conflict of interest statement in the “Confidential to Editor” section, and submit your "Accept" recommendation.

Reviewer #1: (No Response)

Reviewer #2: All comments have been addressed

2. Is the manuscript technically sound, and do the data support the conclusions?

Reviewer #1: Yes

Reviewer #2: Yes

3. Has the statistical analysis been performed appropriately and rigorously? 

Reviewer #1: N/A

Reviewer #2: Yes

4. Have the authors made all data underlying the findings in their manuscript fully available?

Reviewer #1: Yes

Reviewer #2: Yes

5. Is the manuscript presented in an intelligible fashion and written in standard English?

Reviewer #1: Yes

Reviewer #2: Yes

6. Review Comments to the Author

Reviewer #1: The authors have addressed some of my original comments, however a few points remain.

Page 5, lines 7-9 (In relation to my original comment 1): The ‘external validation’ studies (see later comment) still do not fit with the current aims. Perhaps break down the objectives, for example:

1) To identify multivariable prediction models for any outcome at mass gatherings.

2) To summarise evidence for individual predictors of outcomes at mass gatherings

3) To summarise predictive performance of the models in the development studies as well as when applied to new settings.

Measures like R2 would then be for objective 3 and you may wish to summarise performance of models under a separate heading in the results.

Page 6, line 29 & Page 8, line 17: While I appreciate that the authors are considering sample size as part of their risk of bias assessment for studies, I’m still uneasy with the authors saying that they only extracted information for ‘statistically significant’ predictors. If meta-analyses were possible, you wouldn’t exclude non-significant effects as the aim is to pool all the evidence for the predictor so why exclude them here, just because a meta-analysis is not possible in the end? Predictors may appear significant by chance in some studies (overfitting) and not in other studies (when they are important) due to small sample size or because it is highly correlated with another variable in the model. In fact, I think the authors have already addressed this based on another reviewer's comment and now report for all predictors so they could just remove mention of significance in the methods.

External validation studies: I don’t think the authors understood my original comment 8. Because the observations (data points) here are mass gatherings, seeing how well the model predicted for a single mass gathering is just a prediction to a new observation, rather than an external validation as such. It’s the same as predicting risk of death for a single person and comparing to if they died or not – the uncertainty is huge and you would need to predict for many individuals to say whether the model predicts well or not. So here, you would need a dataset with many mass gatherings, similar to what was used to develop the model. The model could over and under predict for individual gatherings but calibrate well overall, so we just don’t know from one or even 3 gatherings. Therefore, I suggest you refer to these predictions as something other than external validation. Maybe refer to it as ‘application of prediction models for mass gatherings’ or something similar and discuss that not much can be concluded from individual predictions but that the models would need to be validated in larger datasets of mass gatherings.

Reviewer #2: (No Response)

7. PLOS authors have the option to publish the peer review history of their article (what does this mean?). If published, this will include your full peer review and any attached files.

Reviewer #1: No

Reviewer #2: Yes: Yohanna Kambai Avong

---

## [Author Response · Author response to Decision Letter 1]

5 Jun 2020

Additional comments from the Editor

This is an interesting manuscript. However, before beinig considered for publication some minor issues in the methods section should be clarified.

1. Please clarify how many reviewers performed the risk of bias assessment. In addition, the risk of bias assessment should be described in an own section. In general, please try to use headings as suggested in PRISMA.

Answer:We revised the manuscript and 1) added that the 2 reviewers independently performed the risk of bias assessment (page 8, lines 26-27), 2) restructured the information of the methods and results section into paragraphs that are now ordered according to the PRISMA checklist (including a separate paragraph on risk of bias assessment).

2. GRADE was not developed to assess studies on risk prediction models. Therefore, the conduct of the GRADE assessment should be described in much more detail.

Answer: We agree with the Editor and further elaborated on using the GRADE approach to rate the certainty of the evidence in prediction modelling studies (see methods section, pages 10-11).

Reviewer #1:

The authors have addressed some of my original comments, however a few points remain.

1. Page 5, lines 7-9 (In relation to my original comment 1): The ‘external validation’ studies (see later comment) still do not fit with the current aims. Perhaps break down the objectives, for example:

1) To identify multivariable prediction models for any outcome at mass gatherings.

2) To summarise evidence for individual predictors of outcomes at mass gatherings.

3) To summarise predictive performance of the models in the development studies as well as when applied to new settings.

Measures like R2 would then be for objective 3 and you may wish to summarise performance of models under a separate heading in the results.

Answer: We agree with the reviewer and rephrased the study objectives at the end of the introduction (Page 5, lines 7-11). In the results section, we rephrased the subheading ‘external model validation studies’ into ‘predictive performance of the models’ and added relevant information (e.g. R2) from the prediction development studies here. (Page 21, lines 12-22).

2. Page 6, line 29 & Page 8, line 17: While I appreciate that the authors are considering sample size as part of their risk of bias assessment for studies, I’m still uneasy with the authors saying that they only extracted information for ‘statistically significant’ predictors. If meta-analyses were possible, you wouldn’t exclude non-significant effects as the aim is to pool all the evidence for the predictor so why exclude them here, just because a meta-analysis is not possible in the end? Predictors may appear significant by chance in some studies (overfitting) and not in other studies (when they are important) due to small sample size or because it is highly correlated with another variable in the model. In fact, I think the authors have already addressed this based on another reviewer's comment and now report for all predictors so they could just remove mention of significance in the methods.

Answer: Correct, we changed this information (i.e. both the statistically significant and non-statistically significant factors from the multivariate models are relevant/important) throughout the manuscript, including at page 6, lines 17-18, and at page 9, line 1.

3. External validation studies: I don’t think the authors understood my original comment 8. Because the observations (data points) here are mass gatherings, seeing how well the model predicted for a single mass gathering is just a prediction to a new observation, rather than an external validation as such. It’s the same as predicting risk of death for a single person and comparing to if they died or not – the uncertainty is huge and you would need to predict for many individuals to say whether the model predicts well or not. So here, you would need a dataset with many mass gatherings, similar to what was used to develop the model. The model could over and under predict for individual gatherings but calibrate well overall, so we just don’t know from one or even 3 gatherings. Therefore, I suggest you refer to these predictions as something other than external validation. Maybe refer to it as ‘application of prediction models for mass gatherings’ or something similar and discuss that not much can be concluded from individual predictions but that the models would need to be validated in larger datasets of mass gatherings.

Answer: We thank the reviewer for this clarification and full agree with this comment. Therefore, we rephrased/removed (where needed) the term validation throughout the manuscript when reporting/discussing on these 3 studies, including that 1) no validation studies of prediction models in big data sets of mass gatherings were identified, 2) applications of prediction models of Arbon/Zeitz in few mass gatherings were present in 3 studies and 3) future validation studies in big data sets is needed.

---

## [Editor Report · Decision Letter 2]

8 Jun 2020

Prediction modelling studies for medical usage rates in mass gatherings: a systematic review

PONE-D-19-28124R2

Dear Dr. Van Remoortel,

We’re pleased to inform you that your manuscript has been judged scientifically suitable for publication and will be formally accepted for publication once it meets all outstanding technical requirements.

Kind regards,

Tim Mathes

Academic Editor

PLOS ONE
---

## [Editor Report · Acceptance letter]

11 Jun 2020

PONE-D-19-28124R2 

Prediction modelling studies for medical usage rates in mass gatherings: a systematic review 

Dear Dr. Van Remoortel:

I'm pleased to inform you that your manuscript has been deemed suitable for publication in PLOS ONE. Congratulations! Your manuscript is now with our production department. 

Kind regards, 

on behalf of

Dr. Tim Mathes 

Academic Editor

PLOS ONE